# Water-soluble vidarabine derivatives alleviate catecholamine-induced heart failure and arrhythmia without impairing cardiac function in mice

Kenji Suita[1], Yoshio Hayakawa[1,2], Yujiro Hoshino[3], Wenqian Cai[4], Reiko Kurotani[5], Yoshiki Ohnuki[1], Yasumasa Mototani[1], Yoshihiro Ishikawa[6], Satoshi Okumura[1]*

**1** Department of Physiology, Tsurumi University School of Dental Medicine, Yokohama, Japan, **2** Department of Dental Anesthesiology, Tsurumi University School of Dental Medicine, Yokohama, Japan, **3** Graduate School of Environment and Information Sciences, Yokohama National University, Yokohama, Japan, **4** Heart Center and Guangzhou Institute of Pediatrics, Guangzhou Women and Children's Medical Center, Guangzhou Medical University, Guangzhou, Guangdong, China, **5** Graduate School of Science and Engineering, Faculty of Engineering, Yamagata University, Yonezawa, Yamagata, Japan, **6** Cardiovascular Research Institute, Yokohama City University Graduate School of Medicine, Kanazawa-ku, Yokohama, Japan

* okumura-s@tsurumi-cu.ac.jp

## Abstract

The worldwide standard guideline for treating heart failure (HF) is inhibition of the patients' chronically enhanced sympathetic nervous system activity. However, despite gains in the treatment of HF with angiotensin and β-adrenergic receptor (β-AR) blockers, some patients do not tolerate β-AR blocking therapy to inhibit cardiac function and contract the respiratory tract. One approach to address this would be adenylyl cyclase (AC) isoform-specific therapy. Indeed, we have demonstrated that vidarabine's selective AC-inhibitory effect in the heart can inhibit the development of HF and arrhythmia without suppressing cardiac function in mice. However, the potential usefulness of vidarabine is limited by its poor solubility, which requires continuous infusion in a large volume of intravenous fluid over a prolonged period. Here, in order to overcome this problem, we aimed to develop vidarabine derivatives with increased solubility and maintained activity. We synthesized derivatives substituted with a (dimethylamino)acetyl group at the 2'-, 3'- or 5'- position of the arabinose ring (V2E, V3E and V5E, respectively) and evaluated their activity *in vitro* and *in vivo*. V2E, V3E and V5E all possess greater water solubility than vidarabine and their inhibitory effect on cardiac AC activity is comparable to that of vidarabine. Furthermore, V2E, V3E and V5E ameliorated the development of HF and reduced the susceptibility to atrial fibrillation in a mouse model.

**Data availability statement:** All relevant data are within the paper and its Supporting information files.

**Funding:** This study was supported by the Japan Society for the Promotion of Science (JSPS) KAKENHI Grant (23K09493 to KS, 23K09517 to YO, 19H03657 to YI, 24K13250 to SO). The founders had no role in study design, data collection and analysis, decision to publish, or preparation of the manuscript.

**Competing interests:** The authors have declared that no competing interests exist.

## Introduction

Heart failure (HF) is one of the leading causes of mortality and hospitalization worldwide [1]. Under physiological conditions, the sympathetic nervous system supports cardiac activity through the modulation of dromotropy, cronotropy, inotropy, and lusinotropy. Moreover, the balance between the sympathetic and parasympathetic nervous systems regulates the peripheral resistance and cardiac output, and plays an essential role in the regulation of cardiovascular function in response to stresses [2]. However, long-term activation of the sympathetic nervous system, characterized by increased release of norepinephrine and epinephrine from both the chromaffin cells of the adrenal gland and heart sympathetic fibers, has a deleterious effect on cardiac structure and performance, leading to progression of HF and arrhythmia [3]. A worldwide standard guideline for treating HF is to inhibit the chronically activated sympathetic nervous system in HF patients, and renin-angiotensin system inhibitors and β-adrenoceptor antagonists (β-blockers) are the main therapies used for this purpose [4].

Controlled clinical trials have shown that β-blockers can reduce the risk of death as well as the risk of hospitalization for cardiovascular causes in patients with HF [5]. However, transient inhibition of cardiac function by β-blockers is a serious obstacle to introduction of the treatment, especially for aged patients. In addition, β-blockers have an inhibitory effect on the respiratory tract, and this can also be a serious problem in aged patients, who often have complications such as pulmonary emphysema [6].

These adverse effects of β-blockers on the respiratory system may be explained by the fact that there are only 3 subtypes ($\beta_1$, $\beta_2$, $\beta_3$) of β-adrenoreceptors (β-AR), and the organ specificity of their expression is relatively low. Since β-ARs are also expressed in the pulmonary bronchus and in cardiac myocytes, β-blockers can cause bronchial smooth muscle contraction and altered cardiac function, resulting in abnormalities of respiratory function and the cardiac circulatory system. β-Blockers are assigned as class II agents in the Vaughan Williams classification, which categorizes antiarrhythmic drugs into class IA, class IB, class IC, class II, class III, class IV and class V according to their mechanism of action [7].

Approximately 30% of cardiac surgical patients develop postoperative atrial fibrillation (POAF) [8]. POAF significantly increases the duration of postoperative hospital stay, hospital costs, and the risk of recurrent AF. Moreover, POAF has been associated with a variety of adverse cardiovascular events such as stroke, HF, and mortality. Perioperative β-blockers are the mainstay for POAF prophylaxis in patients undergoing cardiac surgery, as recommended by international guidelines (Class of Recommendation I, Level of Evidence A in the 2020 European Society of Cardiology [ESC] guidelines for the diagnosis and management of AF) [9]. However, their inhibitory effects on cardiac function and respiration limit the dosage and duration in patients after cardiac surgery. Consequently, there is a need for new β-AR blockade therapy without these adverse effects on cardiac function and respiration.

In this context, it is noteworthy that there are 9 subtypes of adenylyl cyclase (AC), and among them, type 5 AC (AC5) is the major AC isoform in adult cardiac tissue. It is expressed at extremely low levels in other organs, except for the brain [10,11].

Therefore, it seems plausible that selective inhibition of AC5 would have therapeutic effects on HF and arrhythmia, like β-blockers, but without causing adverse effects on cardiac function and the respiratory tract.

We have developed a mouse model with disruption of AC5 [12] and we have also identified the antiviral agent 9-β-D-arabinofuranosyl adenine (vidarabine) as an inhibitor of cardiac AC in mice [6,13]. Building on that work, we found that genetic or pharmacological inhibition of cardiac AC might be associated with resistance to the development of cardiovascular disease (CVD) [10,12] and increased longevity [13,14]. However, the potential usefulness of vidarabine for this purpose is limited by its poor solubility, which results in the need for continuous intravenous administration of a large volume of solution over a prolonged period [15]. In addition, orally administered vidarabine is rapidly metabolized by adenosine deaminase (ADA) in the gastrointestinal tract with loss of its antiviral activity [16]. Consequently, it has to be administered via intravenous infusion [17]. These characteristics hamper the clinical application of vidarabine for the treatment of CVD.

The efficacy of β-blockers on CVD is mainly due to the suppression of cardiac AC activation triggered by the binding of catecholamine to β-AR. Therefore, the direct inhibition of cardiac AC may also prevent the pathological activation of β-AR signaling, resulting in cardiac protection [18]. The nine subtypes of AC differ in their tissue distribution and biochemical properties [11], and the AC5 subtype is expressed most abundantly in adult heart [19]. Our previous study using AC5 knockout mice (AC5KO) revealed that genetic disruption of AC5 did not affect cardiac function at baseline [12]. This result led us to speculate that AC5 inhibition might have a less adverse effect on basal cardiac function than β-AR blockade. In addition, $β_1$-AR, a major cardiac subtype, is also expressed in pulmonary bronchi, so that β-blockers may cause bronchial smooth muscle contraction, leading to asthma attacks [20]. Selective inhibition of cardiac AC5 would be expected to have a less adverse effect than β-blockers on the respiratory function via inhibition of pulmonary β-AR/AC signaling. Consequently, we have been working to develop selective AC5 inhibitors.

Several vidarabine derivatives, in which the 2'-, 3'- or 5'-OH group of the arabinose ring is converted to a more polar functional group, such as phosphate [15], amino acid ester [17], or ester [21], show increased water solubility and resistance to deamination by ADA. More importantly, we revealed that vidarabine possesses a pharmacophore structure for AC5 inhibition at the C2 and C6 positions ($=C^2H-N^1=C^6(NH_2)-$) of the adenine ring [22]. Therefore, modification at the arabinose ring of vidarabine was thought to be a promising approach for overcoming vidarabine's limitations without loss of the AC5-inhibitory effect.

Here, we focused on vidarabine derivatives in which 2'-, 3'- or 5'-OH group of the arabinose ring is substituted by a (dimethylamino)acetyl group, with the aim of increasing the solubility. The newly synthesized vidarabine derivatives were systematically evaluated in comparison with vidarabine for inhibition of cardiac AC enzymatic activity, water solubility, resistance to ADA deamination, adverse effect on cardiac function, cardioprotective effect against catecholamine overload, and antiarrhythmic effect.

## Materials and methods

### Animals

Animal experiments were approved by the Animal Care and Use Committee of Yokohama City University and Tsurumi University School of Dental Medicine. Standard food and water were provided *ad libitum* to mice. 12- to 15-week-old male C57BL/6 wild-type (WT) (Sankyo Labo Service Corporation, Inc., Tokyo, Japan) or AC5KO [12] were used in this study. The previously reported AC5KO mice were repeatedly backcrossed more than 7 times on a C57BL/6 background [10]. Neonatal Wister rats were used for the preparation of primary cell culture in this study (Sankyo Labo Service Corporation, Inc., Tokyo, Japan).

### Ethical approval

All animal experiments complied with the ARRIVE guidelines [23] and were carried out in accordance with the National Institutes of Health guide for the care and use of laboratory animals [24] and institutional guidelines. The experimental protocol was approved by the Animal Care and Use Committee of Tsurumi University (No. 29A041).

## Synthesis of vidarabine derivatives

The vidarabine derivatives (V2E, V3E and V5E) used in the present study were prepared as follows. Proton nuclear magnetic resonance ($^1$H NMR) data are described in the following order: multiplicity (s: singlet, d: doublet, t: triplet, q: quartet and m: multiplet), coupling constant (J) in hertz (Hz) and number of protons.

Disiloxanylidene-protected vidarabine (III) was synthesized from vidarabine (II) (molecular weight (MW): 267.24) as described previously [25]. Reaction with N,N-dimethylglycine in the presence of N,N-dicyclohexylcarbodiimide (DCC) afforded the ester (IV) in high yield. Finally, the protecting group was removed by treatment with tetrabutylammonium fluoride to obtain the 2-substituted derivative of vidarabine, V2E (MW: 352.35) (S4 Fig of S1 Data). Yield: 90%.

1 H NMR (300 MHz, DMSO-$d_6$) δ8.24 (s, 1H), 8.11 (s, 1H), 7.29 (s, 2H), 6.45 (d, J = 6.0 Hz, 1H), 5.84 (d, J = 5.5 Hz, 1H), 5.32 (t, J = 6.0 Hz, 1H), 5.12 (t, J = 5.7 Hz, 1H), 4.46 (q, J = 5.8 Hz, 1H), 3.87–3.61 (m, 3H), 3.05 (d, J = 17.0 Hz, 1H), 2.53 (d, J = 17.2 Hz, 1H), 1.91 (s, 6H). The $^1$H NMR spectrum of V2E is shown in S5 Fig of S1 Data.

Silyl-protected compound (VII) was obtained similarly according to the reported method [17]. Treatment of compound (VII) with N,N-dimethylglycine in the presence of DCC afforded the ester (VIII) in high yield. Finally, the silyl group was removed with tetrabutylammonium fluoride to obtain the 3-substituted product, V3E (Ib) (MW: 352.35) (S6 Fig of S1 Data). Yield: 92%.

1 H NMR (300 MHz, DMSO-$d_6$) δ8.22 (s, 1H), 8.14 (s, 1H), 7.30 (s, 2H), 6.27 (d, J = 4.4 Hz, 1H), 6.07 (s, 1H), 5.28 (t, J = 3.3 Hz, 1H), 5.22 (s, 1H), 4.31 (s, 1H), 4.00–3.96 (m, 1H), 3.70 (s, 2H), 3.29 (s, 2H), 2.28 (s, 6H). The $^1$H NMR spectrum of V3E is shown in S7 Fig of S1 Data.

The diester (XIII) was synthesized from vidarabine (II) according to the previously disclosed method [17]. Treatment of compound (XIII) with N,N-dimethylglycine in the presence of DCC afforded the triester (XIV) in high yield. Finally, the 4-oxopentanoyl group was removed with hydrazine hydrate to obtain the 5-substituted derivative, V5E (Ig) (MW: 352.35) (S8 Fig of S1 Data). Yield: 65%.

1 H NMR (300 MHz, DMSO-$d_6$) δ8.14 (s, 1H), 7.26 (s, 2H), 6.30 (d, J = 4.1 Hz, 1H), 5.79 (d, J = 4.4 Hz, 1H), 5.72 (d, J = 4.1 Hz, 1H), 4.41 (dd, J = 7.1, 11.8 Hz, 1H), 4.30 (dd, J = 3.6, 11.8 Hz, 1H), 4.16 (s, 2H), 3.98 (s, 1H), 3.19 (d, J = 2.8 Hz, 2H), 2.23 (s, 6H) The $^1$H NMR spectrum of V5E is shown in S9 Fig of S1 Data.

## Solubility of V2E, V3E and V5E

Measurement of the water solubility of vidarabine, V2E, V3E and V5E was performed by Absorption Systems (Exton, PA, USA). The equilibrium solubility of vidarabine derivatives was measured in $d$H$_2$O, which was obtained from a Millipore Milli-Q water filtration system. In duplicate, at least ~10 mg of powder of each compound was combined with 0.8 mL of $d$H$_2$O. The samples were shaken on a thermomixer for 24 hours at room temperature. Test compounds vidarabine, V2E, and V3E showed significant residual solid, so the experiment was not continued. However, V5E left no residue. Even after more powder was added to the vials containing V5E, the solution remained clear. The samples were then centrifuged at 14,000 rpm for 10 min. The supernatant of each vial was used to prepare duplicate dilutions, ranging from 100 to 1,000,000-fold, in a 1:1 $d$H$_2$O acetonitrile mixture prior to analysis. All samples were assayed by LC-MS/MS using electrospray ionization against a set of standards prepared in 1:1 $d$H$_2$O:acetonitrile. Standard concentrations ranging from 1.0 µmol/L to 3.0 µmol/L were used. Finally, the equilibrium solubility of each compound was determined to be as follows: vidarabine 0.46 mg/mL, V2E 1.14 mg/mL (2.48-fold relative to vidarabine), V3E 1.56 mg/mL (3.39-fold relative to vidarabine), and V5E > 18.4 mg/mL (> 40.0-fold relative to vidarabine).

## Drug treatment

V2E, V3E and V5E synthesized by Chemgenesis Inc. (Tokyo, Japan) were used in this study. Other reagents were purchased from Sigma Aldrich (St. Louis, MO, USA) unless otherwise indicated. Based on our previous report [6], the

vidarabine (Wako Pure Chemical Industries, Osaka, Japan) dose was set to 15 mg/kg, which is available as the maximal daily dose in the treatment of herpes simplex encephalitis [26]. The doses of V2E, V3E and V5E were set as equimolar with that of vidarabine (approximately 19.7 mg/kg per day).

The dose of vidarabine (15 mg/kg/day; a dose approved for clinical use in humans) was selected based upon that used in previous studies: this dose did not eliminate the inotropic effects of acute isoproterenol (ISO), did not depress cardiac function at baseline, and retained high selectivity for AC5 [13]. The doses of V2E, V3E and V5E used in this study were set to be equimolar with that of vidarabine. In physiological experiments, mice were administered vidarabine, V2E, V3E or V5E for 6 days via subcutaneously implanted osmotic mini-pumps (Alzet model 2001; Cupertino, CA, USA) unless otherwise stated. Alzet osmotic mini-pumps are small, implantable pumps used for research in mice, rats, and other laboratory animals, enabling accurate and continuous dosing of unrestrained laboratory animals.

For sympathetic activation, ISO or saline (Otsuka Pharmaceutical, Tokyo, Japan) as a vehicle was administered to mice. Long-term infusion of ISO was performed for 7 days at a dose of 60 mg/kg/day with an osmotic mini-pump as described previously [10]. The ISO pumps were removed 24 hours before biochemical and physiological studies.

## AC assay

AC assay using membrane preparation from cardiac and lung tissues was performed as described previously with some modifications [6]. Mice were sacrificed by cervical dislocation. Tissues were homogenized in buffer A consisting of 50 mmol/L Tris-HCl (pH 8.0), 2 mmol/L EGTA, 10 μmol/L phenylmethylsulfonyl fluoride (PMSF), 10 μg/mL leupeptine, and 50 U/mL egg-white trypsin inhibitor (ETI). The homogenate was centrifuged at $500 \times g$ for 15 min at 4°C, then at 50,000 x g for 30 min at 4°C. The pellet was resuspended in buffer B containing 50 mmol/L Tris-HCl (pH 8.0), 1 mmol/L EGTA, 1 μmol/L PMSF, 5 μg/ mL leupeptin, and 5 U/mL ETI. The AC assay with 50 μmol/L. forskolin or 10 μmol/L ISO was performed in reaction solution (25 mmol/L HEPES (pH 8.0), 5 mmol/L $MgCl_2$, 0.5 mmol/L EDTA, 0.1 mmol/L ATP, 1 mmol/L creatine phosphate, 8 U/mL creatine phosphokinase, 0.2 mmol/L isobutylmethylxanthine) containing several concentrations of vidarabine, V2E, V3E or V5E. For ADA treatment, 100 μmol/L vidarabine, V2E, V3E and V5E was preincubated with ADA (0.08, 0.32, 1.25, 5 U/mL) at 27°C for 5 min [27,28]. After addition of the membrane proteins, the reaction mixture was incubated at 30°C for 15 min. The reaction was stopped by adding an equal volume of ice-cold 20% (volume/volume, v/v) trichloroacetic acid (TCA). The solution was incubated on ice for 60 min, then centrifuged at $13,000 \times g$ for 10 min at 4°C. The supernatant of the reaction mixture was subjected to radioimmunoassay using [$I^{125}$]cAMP (PerkinElmer, Waltham, MA, USA) to determine cAMP levels.

## Assessment of cardiac function

Mice were anesthetized by isoflurane (Mylan, Canonsburg, PA, USA) inhalation to maintain the lightest anesthesia possible. On average, 1.0–1.5% (v/v) isoflurane was required for adequate anesthesia under our conditions. Cardiac function was measured by means of echocardiography according to our previous reports [18,29]. We assessed left ventricular (LV) diastolic diameter, LV systolic diameter, EF and %FS. %FS was calculated by use of the following formula: %FS = 100 x (diastolic LV diameter – systolic LV diameter)/diastolic LV diameter).

## Induction of AF

Paroxysmal AF was induced by transesophageal atrial burst pacing in mice according to the method previously described, with minor modifications [30]. Under isoflurane anesthesia (1.5% − 2.0% for maintenance), a 1.1 French octa-polar catheter (EPR800; Millar Instruments, Houston, TX, USA) was carefully inserted into the esophagus of each mouse. The catheter was fixed where the height of the atrial electrogram was highest on the esophageal electrocardiogram (ECG). Ten minutes after intraperitoneal administration of 1.5 mg/kg ISO (the same conditions as those used for norepinephrine

[6,30], transesophageal atrial burst pacing was conducted for 10 s at a stimulation amplitude of 1.5 mA with 10-ms cycle lengths and a pulse width of 3 ms. The duration of AF was measured based on its apparent length on the lead II body surface ECG, with AF on ECG defined by the following criteria as fellows: i) loss of P wave, ii) irregular R-R interval, iii) duration longer than 2 s [31]. Three trials of burst pacing were performed in individual mice, and the longest episode was taken as the duration of AF [31].

## Measurement of fibrosis area in heart tissue

Heart specimens were fixed with formalin, embedded in paraffin, and sectioned at 6-μm thickness. Masson-trichrome staining using a Trichrome Stain Kit (HT15, Sigma-Aldrich) was conducted in accordance with the manufacturer's instructions. In Masson-trichrome staining, fibers stained in aniline blue were collagen fibers, while those stained in red were muscle fibers. Sections of the cardiac tissues were outlined manually to define regions of interest (ROIs). We measured the percentage fibrosis within 3–5 ROIs for each section, using Image J 1.48v software (National Institute of Health, Bethesda, MD, USA) [32,33].

## Primary cultures of neonatal rat cardiac myocytes

Primary cultures of rat neonatal cardiomyocytes (NCMs) were prepared by the method originally described by Simpson and Savion with minor modifications [34]. Briefly, the hearts from 1- to 3-day-old Wister rats (Japan SLC Inc., Hamamatsu, Japan) were minced and treated with 0.04% type II collagenase (Sigma) and 0.04% pancreatin (Nacalai Tesque, Kyoto, Japan). The dispersed cells were collected and seeded into gelation-coated dishes at $5 \times 10^5$ cells/35 mm dish or $1.2 \times 10^5$ cell/24 mm dish. Cells were incubated in Dolbecco's modified Eagle's medium (DMEM/F12) supplemented with 5% horse serum and 10μmol/L cytosine arabinoside (AraC), which preferentially reduces the proportion of proliferative non-myocytes. Eighteen hours later, the culture medium was exchanged to DMEM/F12 including insulin-transferrin-selenium-A supplement (ITS-A, Invitrogen), 100 U/mL penicillian, 100 μg/mL streptomycin, and 10 mmol/L glutamine. The cells were plated on plastic dishes coated with 1% (w/v) gelatin. Twenty-four hours after seeding, the culture medium was changed to serum-free low-grade DMEM and myocytes were incubated for 24 hours after seeding under a humidified 5% $CO_2$, 95% air incubator at 37°C.

## Evaluation of cardiomyocyte apoptosis

Evaluation of apoptosis of ISO-induced NCMs [35] by *in situ* labeling of fragmented DNA in cardiomyocytes was performed using the DeadEnd™ fluorometric TUNEL system (Promega, Madison, WI, USA) based on the previously described protocol with some modifications [36]. Cardiac myocytes were incubated with 100 μmol/L ISO in the presence of 100 μmol/L vidarabine, V2E, V3E or V5E for 48 hours. The number of terminal deoxyribonucleotidyl transferase (TdT)-mediated biotin-16-deoxyuridine triphosphate (dUTP) nick-end labeling (TUNEL)-positive nuclei was counted manually in the sections from each group over a microscopic field of 20 x in a blinded manner. The percentage of TUNEL-positive nuclei relative to the total number of myocytes, indicated by DAPI (4',6-diamidino-2 phenylindole)-stained nuclei, in the same field was calculated.

## Western blotting

ISO (60 mg/kg/day) with vidarabine (15 mg/kg/day) or V2E (19.7 mg/kg/day) was administered to WT mice via osmotic mini-pumps for 7 days. The excised ventricles were homogenized in ice-cold RIPA buffer with proteinase inhibitor cocktail (Thermo Fisher Scientific, Waltham, MA, USA) in a Polytron® (Kinematica AG, Lucerne, Switzerland), and the homogenate was centrifuged at $13,000 \times g$ for 10 min at 4°C. The supernatant was collected, and the protein concentration was measured using a DC protein assay kit (Bio-Rad, Hercules, CA, USA). Equal amounts of protein were subjected to SDS-polyacrylamide gel electrophoresis and blotted onto PVDF membrane (Millipore, Billerica, MA, USA). The membranes were incubated overnight at 4°C with primary antibodies. Primary antibody against calpain 1 (1:500, #ab108400) was

purchased from Abcam (Cambridge, UK). Primary antibodies against ryanodine receptor (RyR2) (1:1000, #MA3–916) and phospho-RyR2 at serine-2808 (1:2500, #A010-30AP) were purchased from Thermo Fisher Scientific and Badrilla (Leeds, UK), respectively. After incubation with primary antibody, membranes were incubated with horseradish peroxidase-conjugated secondary antibody at room temperature for one hour. Blots were visualized by chemiluminescence (Cytiva/Global Life Sciences Solutions, Marlborough, MA, USA), and the density of signals was quantified using ImageJ software.

### Statistical analyses

Data are presented as means ± standard deviation in this study. Student's *t* test was conducted for the statistical analysis of two groups (S1A-B Fig, S3B Fig and S10A-C Fig of S1 Data). For multiple comparisons, one-way analysis of variance (ANOVA) followed by the Tukey-Kramer's *post hoc* test (Figs 5B–7B, S11 Fig and S12A-B Fig of S1 Data) or two-way ANOVA followed by Bonferroni's *post hoc* test (Figs 2A–D–4A, B, S2 Fig and S3A Fig of S1 Data) was performed. A P value less than 0.05 was considered statistically significant. All raw data can be found in S3 Data.

### Results

#### V2E, V3E and V5E are more water soluble than vidarabine

Previously, it was reported that substitution of vidarabine's arabinoside ring with 5'-phosphate ester, 5'-amino acid ester and 2',3'-diacyl ester increases water solubility and ADA resistance, while retaining high antiviral activity [15,17]. We have shown that classic P-site inhibitors, which are adenosine analogs that inhibit AC catalytic activity in a non- or uncompetitive manner with respect to ATP [37], possess a pharmacophore structure at the C2 and C6 positions (=$C^2H-N^1=C^6(NH_2)$-) of the adenine ring for both AC-inhibitory effect and AC5 selectivity (Fig 1A) [22]. We thus synthesized three types of vidarabine derivatives substituted with a (dimethylamino)acetic acid group at the 2'-, 3'- and 5'- position of arabinose ring (V2E, V3E and V5E, respectively) (Fig 1B).

We first compared the water solubility of V2E, V3E and V5E with that of vidarabine. The solubility of vidarabine in water was 0.46 mg/mL, in agreement with a previous report [38]. The water solubility of both V2E and V3E was increased (2.48-fold and 3.39-fold relative to vidarabine, respectively), and that of V5E was greatly increased (> 18.4-fold) (Table 1).

#### V2E, V3E and V5E inhibit AC5 with greater specificity than vidarabine

We next compared the inhibitory effects of vidarabine, V2E, V3E and V5E on ISO-stimulated AC activity (100 μmol/L) using cardiac membrane prepared from WT and AC5KO mice. Ventricular AC activity was significantly decreased in AC5KO relative to WT mice (WT: 259 ± 23 pmol cAMP/mg/min vs AC5KO: 185 ± 14 pmol cAMP/mg/min, *P* < 0.01) (S1A Fig of S1 Data), whereas lung AC activity was similar in WT and AC5KO mice (WT: 141 ± 13 pmol cAMP/mg/min vs AC5KO: 138 ± 19 pmol cAMP/mg/min) (S1B Fig of S1 Data). V2E, V3E and V5E decreased AC activity in WT in a dose-dependent manner (Fig 2A), like vidarabine. More importantly, V2E, V3E and V5E did not decrease ventricular AC activity in AC5KO relative to vidarabine (Fig 2B).

We also compared the inhibitory effects of vidarabine, V2E, V3E and V5E on forskolin-stimulated AC activity (50 μmol/L) using lung membrane prepared from WT and AC5KO mice. We found that V3E decreased AC activity similarly to vidarabine. However, V2E and V5E did not show inhibitory effects on forskolin-stimulated AC activity at low concentration (1 μmol/L and 10 μmol/L) (Fig 2C, D).

These data suggest that V2E, V3E and V5E selectively inhibit AC5, compared to other AC subtypes expressed in the heart of AC5KO (Fig 2A, B, Table 1).

#### V2E, V3E and V5E inhibit AC5 more potently than AC6 in the heart

We have previously demonstrated that there is no compensatory increase in the other minor cardiac AC subtypes (AC2, AC3, AC4, AC7, AC9) in AC5KO [12], and AC6 might be the predominant isoform in AC5KO heart. We thus examined the

**A**

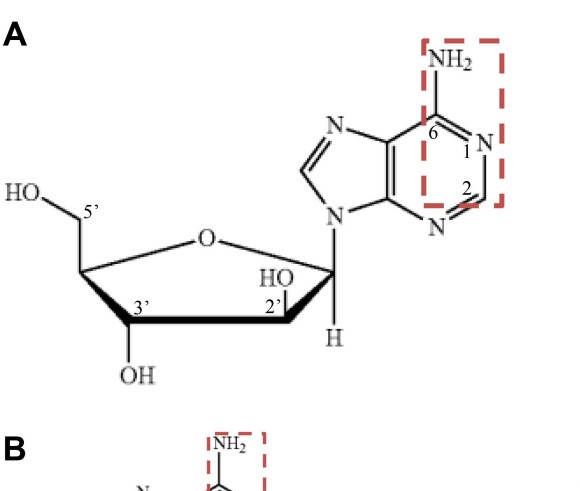

**B**

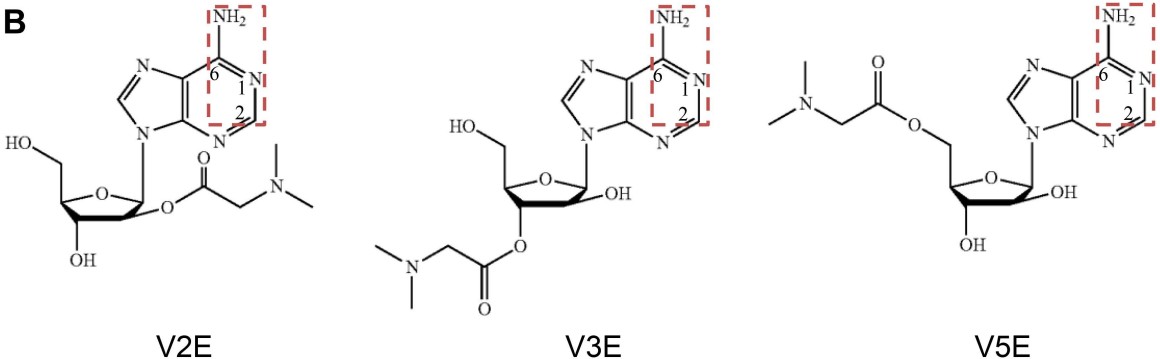

V2E                           V3E                           V5E

**Fig 1. Structures of vidarabine, V2E, V3E and V5E.** Chemical structures of vidarabine (A) and vidarabine derivatives (V2E, V3E and V5E) (B). The essential pharmacophore structure ($=C^2H-N^1=C^6(NH_2)-$) for selective inhibition of AC5 is framed with a red-dotted rectangle.

**Table 1. Summary of the key characteristics of V2E, V3E and V5E.**

|     | Water solubility | AC5 selectivity[a] | Resistance to ADA[b] | Inhibition of cardiac dysfunction[c] | Inhibition of AF[d] |
|-----|------------------|--------------------|----------------------|--------------------------------------|---------------------|
| V2E | +                | +                  | +                    | *                                    | *                   |
| V3E | +                | +                  | +++                  | *                                    | *                   |
| V5E | +++              | +                  | +                    | +                                    | *                   |

[a]Data from Fig 2.

[b]Data from Fig 3.

[c]Data from Fig 4.

[d]Data from Fig 7.

+: The parameter was greater than vidarabine.

+++: The parameter was much greater than vidarabine.

*: The parameter was comparable with vidarabine.

effects of V2E, V3E and V5E on the ISO-stimulated AC activity (50 µM) in the heart of WT and AC5KO to investigate the inhibitory potency towards AC6. The inhibitory effects of V2E (S10A Fig of S1 Data), V3E (S10B Fig of S1 Data) and V5E (S10C Fig of S1 Data) on ISO (50 µM)-stimulated AC activity in the heart of AC5KO were significantly smaller than in the case of WT mice (V2E: WT 87.7±5.9% vs AC5KO 94.7±1.7%, $P<0.05$; V3E: WT 83.9±7.7% vs AC5KO 94.3±2.1%, $P<0.05$; V5E: WT 89.5±5.9% vs AC5KO 98.5±5.0%, $P<0.05$).

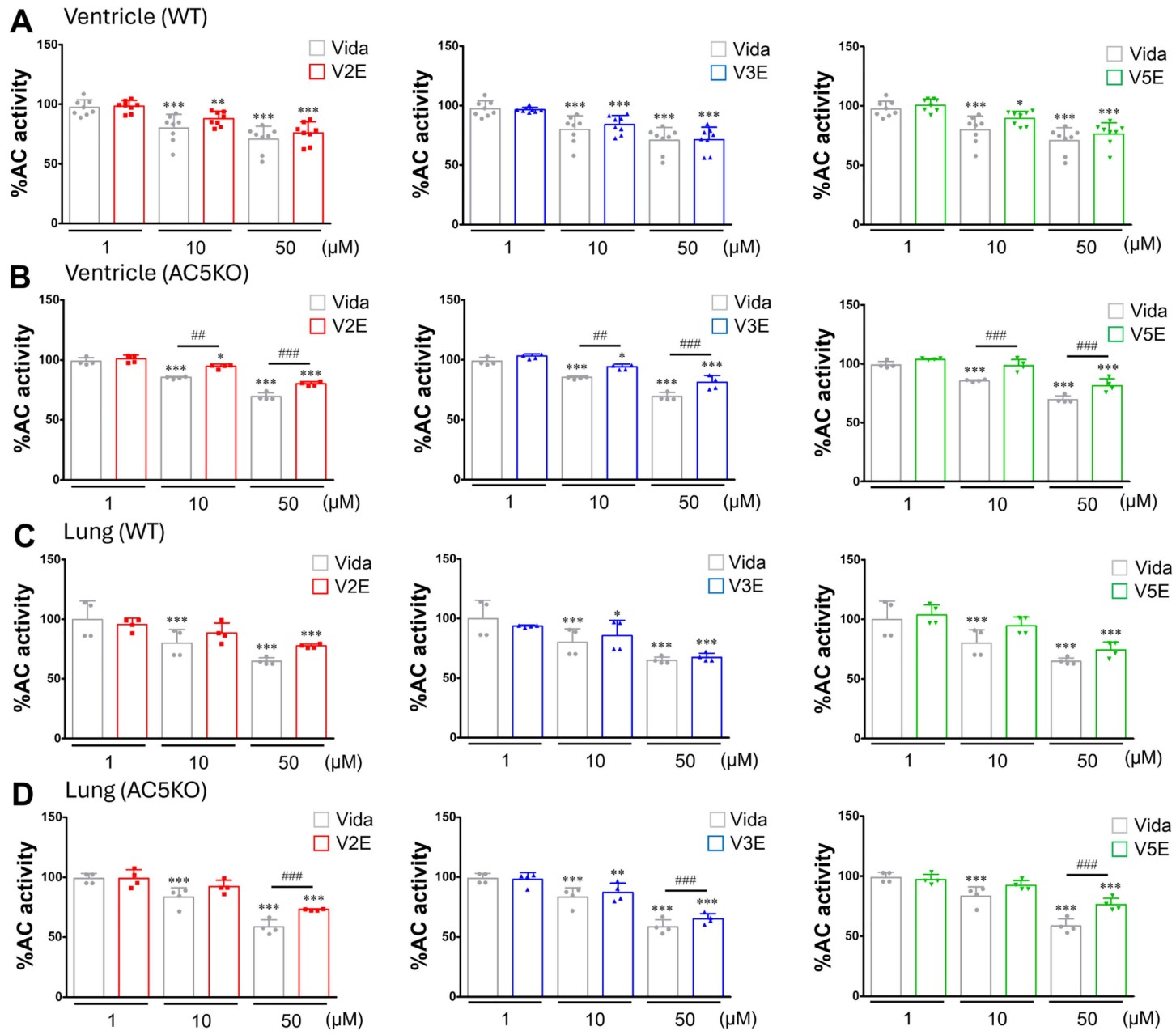

**Fig 2. Effects of vidarabine, V2E, V3E and V5E on AC activity in the heart of WT and AC5KO mice.** (A) Vidarabine (Vida), V2E, V3E and V5E similarly and significantly decreased ISO (50 µmol/L)-stimulated AC activity in the WT heart in a dose-dependent manner. (B) Inhibitory effects of V2E, V3E and V5E on ISO (50 µmol/L)-stimulated AC activity were significantly smaller than those of vidarabine at 10 µmol/L and 50µmol/L concentration of of each compound in the AC5KO heart. (C) Vida and V3E significantly decreased forskolin (50 µmol/L) -stimulated AC activity, but neither V2E nor V5E had any effect at 10 µmol/L concentration in WT lung. However, all of them significantly decreased forskolin-stimulated AC activity at 50 µmol/L. (D) Inhibitory effects of V2E, V3E and V5E on forskolin (50 µmol/L)-stimulated AC activity were significantly smaller than that of vidarabine at 50 µmol/L concentration of each compound in the AC5KO lung. The AC activity without inhibitors (Vida, V2E, V3E and V5E each) was set to 100 in each case. $^{*}P < 0.05$, $^{**}P < 0.01$, $^{***}P < 0.001$ vs control (without inhibitor); $^{\#\#}P < 0.01$, $^{\#\#\#}P < 0.001$ vs Vida (at each concentration) by two-way ANOVA with Bonferroni's *post hoc* test. Data are presented as mean ± SD and dots show individual data from eight WT mice and four AC5KO mice, respectively (**A**: $n = 8$, **B**: $n = 4$, **C**: $n = 4$, **D**: $n = 4$).

These data suggests that V2E, V3E and V5E are less potent inhibitors of AC6 in the heart, as compared with AC5, as in the case with vidarabine [13].

## V3E is more resistant than vidarabine to deamination by ADA

Vidarabine is rapidly deaminated by ADA to yield 9-β-D-arabinofuranosyl hypoxanthine (Ara-H), resulting in loss of anti-viral activity [16,21]. Thus, we treated 100 µmol/L of vidarabine, V2E, V3E or V5E with various concentrations of ADA (0, 0.08, 0.32, 1.25 and 5 U/mL) for 5 min and examined the inhibitory effects on ventricular AC activity (Fig 3A–C). ADA concentration-dependently abrogated the inhibitory effects of these compounds on forskolin-stimulated AC activity (50 µmol/L).

These data suggest that Ara-H has little or no inhibitory effect on AC activity. In contrast, V3E still significantly suppressed ventricular AC activity at various ADA concentrations (0.32, 1.25 U/mL) at which vidarabine completely lacked inhibitory effect (Fig 3B, Table 1).

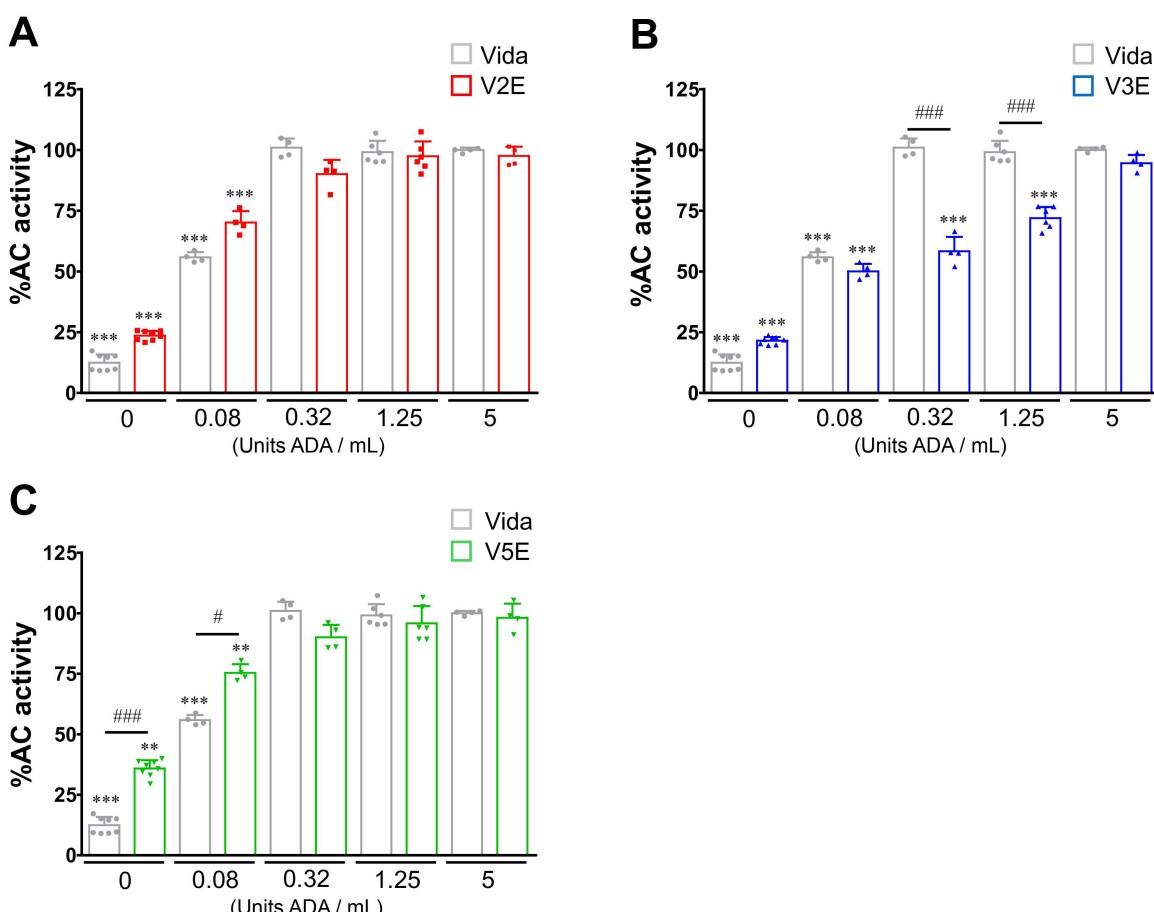

**Fig 3. Effects of ADA on the AC-inhibitory activity of vidarabine, V2E, V3E and V5E in the heart.** The effects of vidarabine (Vida), V2E, V3E and V5E on forskolin (50 µmol/L)-stimulated AC activity were inhibited by ADA in a concentration-dependent manner. V3E was most resistant to ADA-mediated deamination. The AC activity without inhibitor (Vida, V2E, V3E or V5E) was set to 100 in each case. $^{**}P < 0.01$, $^{***}P < 0.001$ vs control (without inhibitor); $^{##}P < 0.05$, $^{###}P < 0.001$ vs Vida (at each concentration) by two-way ANOVA with Bonferroni's *post hoc* test. Data are presented as mean ± SD and dots show individual data from three independent experiments performed with membrane preparations of four WT mice ($n = 4$ - 8).

## V2E, V3E and V5E decrease cardiac dysfunction induced by chronic ISO infusion

We compared the protective effects of V2E, V3E and V5E on cardiac function with that of vidarabine using an experimental HF model induced by chronic ISO infusion at a dose of 60 mg/kg/day for 1 week (Fig 4) [10]. Vidarabine (15 mg/kg/day), V2E (19.7 mg/kg/day), V3E (19.7 mg/kg/day) and V5E (19.7 mg/kg/day) did not alter the left ventricular ejection fraction (EF) (Fig 4A), %fractional shortening (%FS) (Fig 4B) or heart rate (HR) (S2 Fig of S1 Data) before ISO infusion. However, EF and %FS were significantly decreased after chronic ISO in the control WT mice (vehicle vs ISO: EF: 72 ± 0.5% vs 61 ± 1.8%, *P* < 0.001; %FS: 34 ± 0.4% vs 30 ± 1.2%, *P* < 0.001). This is in agreement with previous findings [10,13]. However, the decrease of EF was significantly inhibited in V2E-, V3E- and V5E-treated mice, compared to the control after chronic ISO, as in the case of vidarabine [13] (vidarabine: 65 ± 1.7%, *P* < 0.001 vs control; V2E 66 ± 1.6%, *P* < 0.001 vs control; V3E 66 ± 0.5%, *P* < 0.001 vs control; V5E 69 ± 2.4%, *P* < 0.001 vs control). In particular, the V5E-treated group showed increased cardiac function, compared to the vidarabine-treated group (*P* < 0.01). The %FS showed a similar tendency to EF (Fig 4, Table 1). On the other hand, HR was similar among the five groups before (*left*) and after (*right*) chronic ISO infusion (S2 Fig of S1 Data).

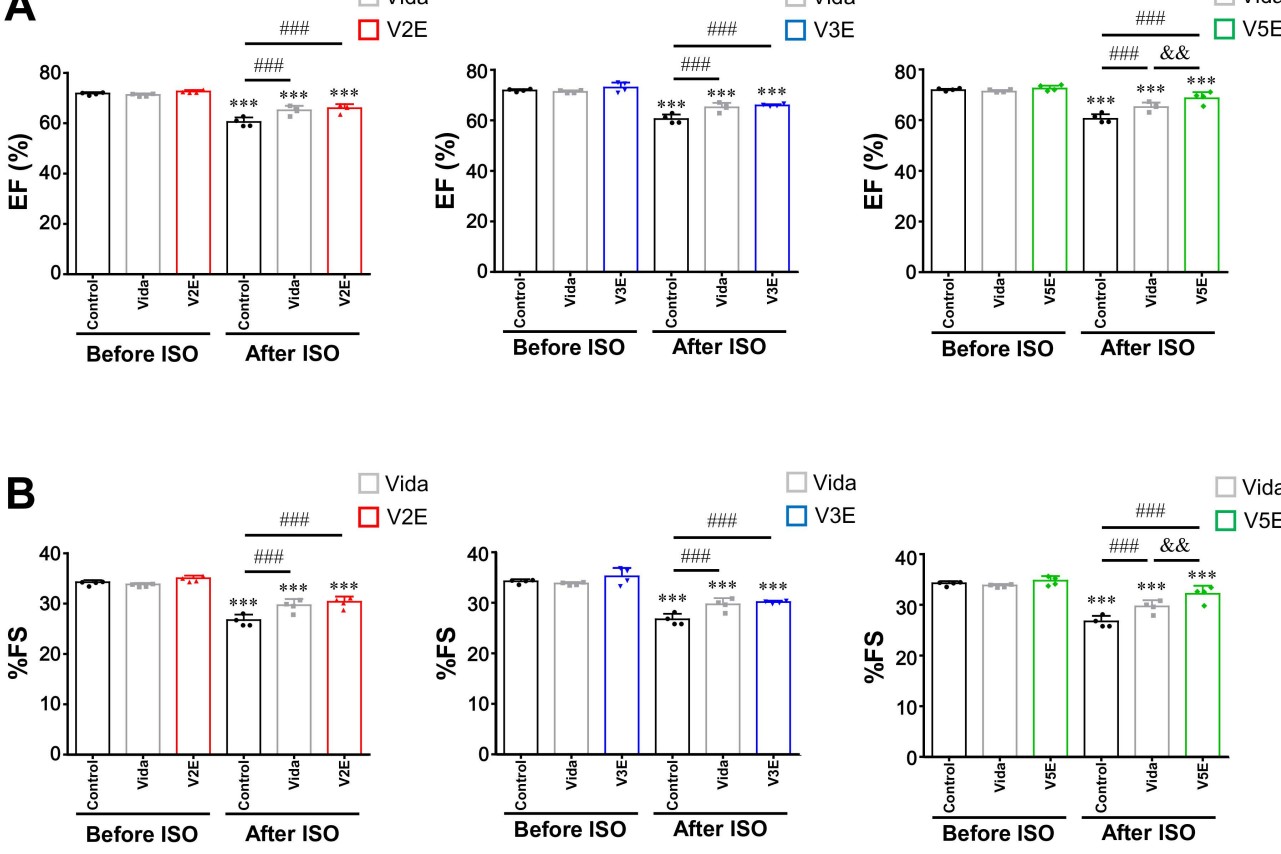

**Fig 4. Effects of vidarabine, V2E, V3E and V5E on cardiac dysfunction after chronic ISO infusion.** Echocardiography was performed to examine the effects of vidarabine (Vida), V2E, V3E and V5E on cardiac function in terms of EF (A) and %FS (B) before (*left*) and after (*right*) chronic ISO infusion. **P* < 0.01, ***P* < 0.001 vs before ISO (without inhibitor in each case); ###*P* < 0.001 vs control (after ISO) and &&*P* < 0.01 vs Vida (after ISO) by two-way ANOVA with Bonferroni's *post hoc* test. Data are presented as mean ± SD and dots show individual data from four WT mice (*n* = 4).

## V2E, V3E and V5E ameliorate cardiac fibrosis after chronic ISO

We examined the effects of vidarabine, V2E, V3E, and V5E on cardiac fibrosis after chronic ISO treatment (Fig 5A) [10]. Cardiac fibrosis was significantly increased after chronic ISO treatment for 1 week (control vs ISO: 0.7±0.2% vs 1.3±0.2%, $P<0.01$), as demonstrated previously [13]. However, V2E, V3E and V5E significantly decreased ISO-induced cardiac fibrosis (Fig 5B). More importantly, we confirmed that the protective effects on ISO-induced cardiac fibrosis were comparable to that of vidarabine (ISO+vidarabine: 0.5±0.1%, $P<0.001$ vs ISO; ISO+V2E: 0.6±0.2%, $P<0.01$ vs ISO; ISO+V3E: 0.6±0.3%, $P<0.01$ vs ISO; ISO+V5E: 0.7±0.1%, $P<0.01$ vs ISO) [13] (Fig 5B).

## V2E, V3E and V5E ameliorate apoptosis of cardiac myocytes after chronic ISO

We next examined the effects of vidarabine, V2E, V3E, and V5E on apoptosis of NCMs after chronic ISO treatment by means of TUNEL (Fig 6A). Apoptosis was significantly increased after chronic ISO treatment for 48 hours (control vs ISO: 5.7±0.6% vs 12.1±2.2%, $P<0.001$), in accordance with previous findings [13]. V2E, V3E and V5E significantly decreased ISO-promoted apoptosis (Fig 6B). Furthermore, we confirmed that the protective effects against ISO-promoted apoptosis of cardiac myocytes were comparable to that of vidarabine [13] (ISO+vidarabine: 8.3±0.6%, $P<0.01$ vs ISO; ISO+V2E: 8.1±0.9%, $P<0.001$ vs ISO; ISO+V3E: 8.2±0.7%, $P<0.001$ vs ISO; ISO+V5E: 7.9±1.0%, $P<0.001$ vs ISO).

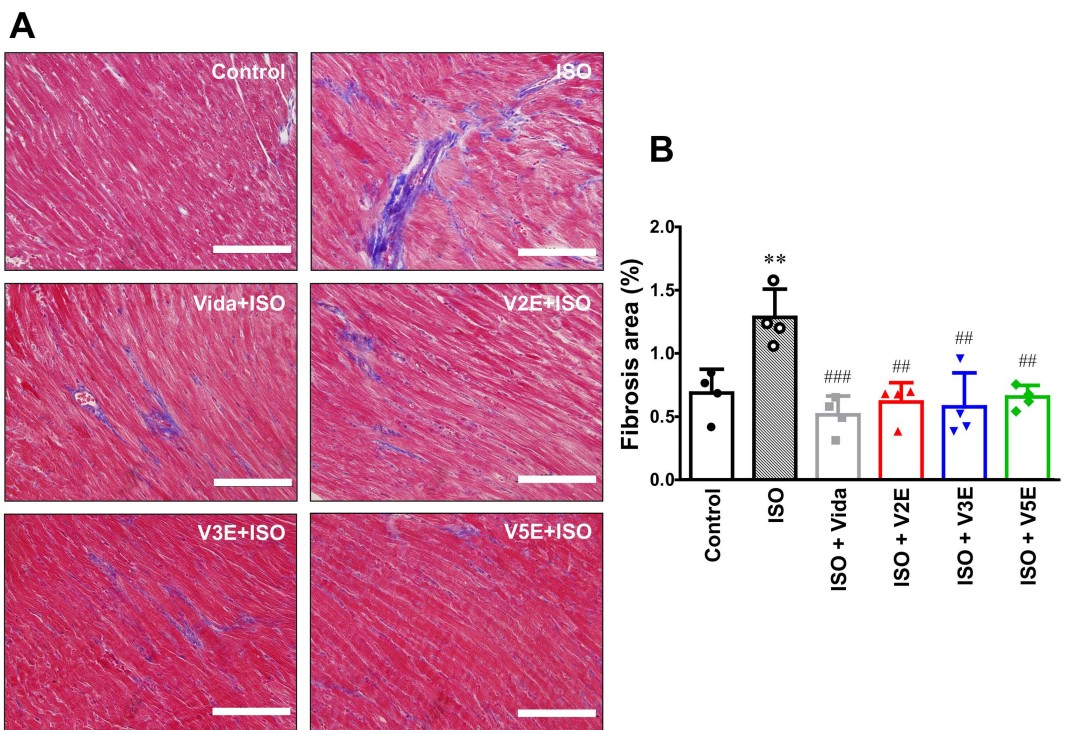

**Fig 5. Effects of vidarabine, V2E, V3E and V5E on cardiac fibrosis.** We examined the effects of chronic ISO infusion (60 mg/kg/day for 1 week) via osmotic mini-pumps in WT mice with/without vidarabine (Vida), V2E, V3E or V5E on cardiac fibrosis by means of Masson-trichrome staining. (A) Representative images of Masson-trichrome-stained ventricular muscle sections of WT mice treated with ISO (60 mg/kg/day) plus Vida (15 mg/kg/day), V2E (19.7 mg/kg/day), V3E (19.7 mg/kg/day) or V5E (19.7 mg/kg/day) for 7 days. Scale bar: 100 μm. (B) Chronic ISO infusion significantly increased the area of fibrosis in cardiac muscle but this increase was significantly decreased by Vida, V2E, V3E and V5E. Statistical significance was analyzed by one-way ANOVA with the Tukey-Kramer's *post hoc* test. **$P<0.01$ vs control (without inhibitor); ##$P<0.01$ vs ISO. Data are presented as mean±SD and dots show individual data from four WT mice ($n=4$).

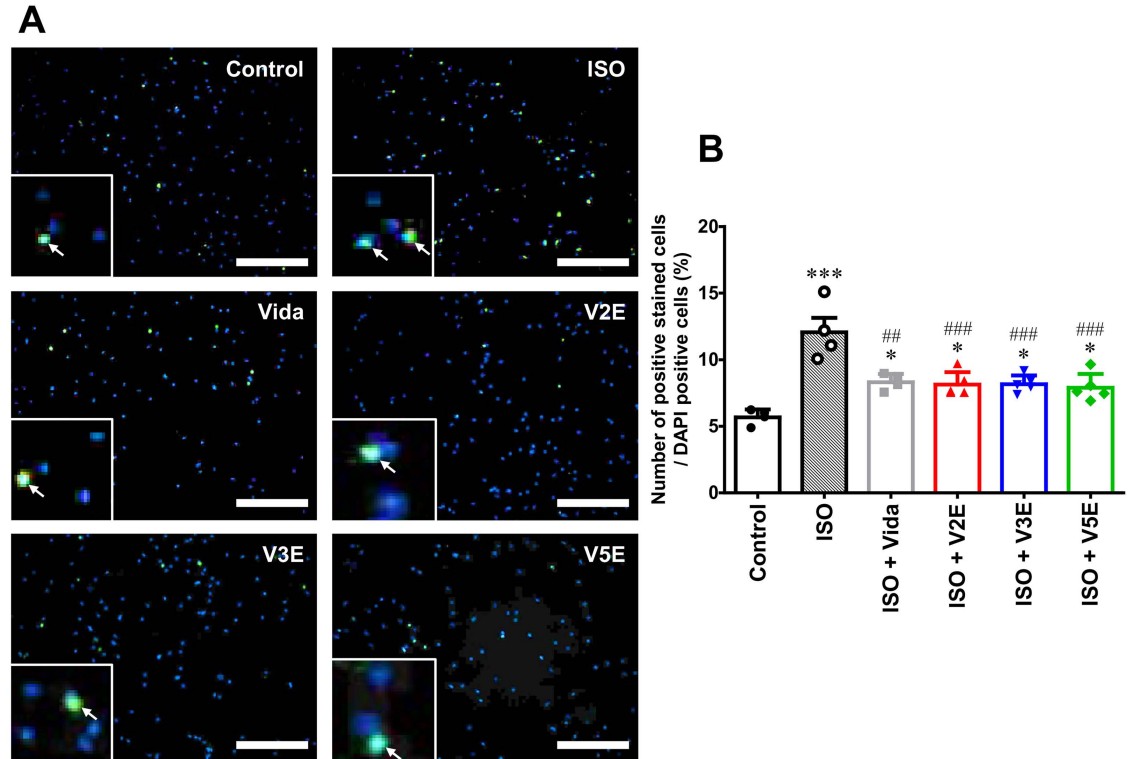

**Fig 6. Effects of vidarabine, V2E, V3E and V5E on apoptosis of cardiac myocytes.** (A) The effects of vidarabine (Vida), V2E, V3E and V5E on cultured NCMs were evaluated by means of TUNEL staining 48 hours after treatment of NCMs with ISO (100 μM) in the presence of Vida, V2E, V3E or V5E respectively (100 μM). The inset show 4x maginified images. Arrows in insets indicate TUNEL-positive nuclei. Scale bar: 100 μm. (B) The percentage of TUNEL-positive nuclei (*green*) relative to the total number of total DAPI-stained nuclei (*blue*) is shown as a bar graph. Statistical significance was analyzed by one-way ANOVA with the Tukey-Kramer *post hoc* test. ***$P < 0.001$ vs control (without inhibitor); ##$P < 0.01$, ###$P < 0.001$ vs ISO. Data are presented as mean±SD and dots show individual data from two independent experiments with similar results (Control: $n = 4$, ISO: $n = 4$, ISO+Vida: $n = 4$, ISO+V2E: $n = 5$, ISO+V3E: $n = 5$, ISO+V5E: $n = 5$).

## V2E ameliorates cardiac hypertrophy after chronic ISO

We examined the effects of vidarabine and V2E on cardiac hypertrophy after chronic ISO treatment (S11 Fig of S1 Data). Cardiac hypertrophy as exemplified by ventricular weight per body weight ratio (mg/g) was significantly increased after chronic ISO treatment for 1 week (control vs. ISO: 4.3±0.3 mg/g vs. 5.2±0.2 mg/g, $P < 0.01$), in accordance with previous findings [18]. However, vidarabine and V2E significantly decreased ISO-induced cardiac hypertrophy (ISO+vidarabine: 4.5±0.1 mg/g, $P < 0.001$ vs. ISO; ISO+V2E: 4.7±0.2 mg/g, $P < 0.01$ vs. ISO) (S11 Fig of S1 Data).

These data suggest that vidarabine and V2E might ameliorate cardiac hypertrophy after chronic ISO to comparable extents.

## Disruption of AC5 shortens the duration of catecholamine-elongated AF

AF is the most prevalent cardiac arrhythmia, especially among elderly people, and is a source of considerable morbidity and mortality [39]. Numerous animal and human studies have demonstrated that the autonomic imbalance is closely associated with the initiation and maintenance of AF [40,41].

We first examined the effects of *AC5*-gene deficiency on AF duration in a model of AF induction by transesophageal atrial burst pacing with excessive sympathetic activation, as previously described by us [30]. Sympathetic stimulation by

ISO injection strikingly elongated the AF duration in WT mice (Control: 27.5±0.6 sec vs ISO: 966±465 sec, $P<0.001$). However, the elongation was significantly suppressed in AC5KO mice (ISO: 335±274 sec, $P<0.01$ vs ISO in wild type) (S3A Fig of S1 Data). Atrial AC activity was significantly reduced in AC5KO mice compared to that of WT mice (WT: 278±34.5 pmol cAMP/mg/min vs AC5KO: 160±17.2 pmol cAMP/mg/min, $P<0.01$) (S3B Fig of S1 Data). This result suggests that a reduction of atrial AC5 activity might contribute, at least in part, to the suppression of AF.

## V2E, V3E and V5E shorten the duration of catecholamine-elongated AF

We next examined the effects of vidarabine, V2E, V3E and V5E on AF duration in the transesophageal atrial burst pacing model in mice (Fig 7A), because suppressing the activity of AC5, and not the entire β-adrenergic signaling pathway, may be preferable to current β-AR blockade therapy for the treatment of HF and arrhythmia. Specifically, it is expected to reduce the risk of adverse effects on cardiac function and the respiratory tract, especially in aged patients.

Administration of vidarabine, V2E, V3E and V5E was performed with osmotic mini-pumps for 6 days. Vidarabine, V2E, V3E and V5E similarly and significantly decreased ISO-induced AF duration. More importantly, we confirmed that the inhibitory effects of V2E, V3E and V5E on ISO-induced AF duration were comparable to that of vidarabine (Fig 7B, C, Table 1).

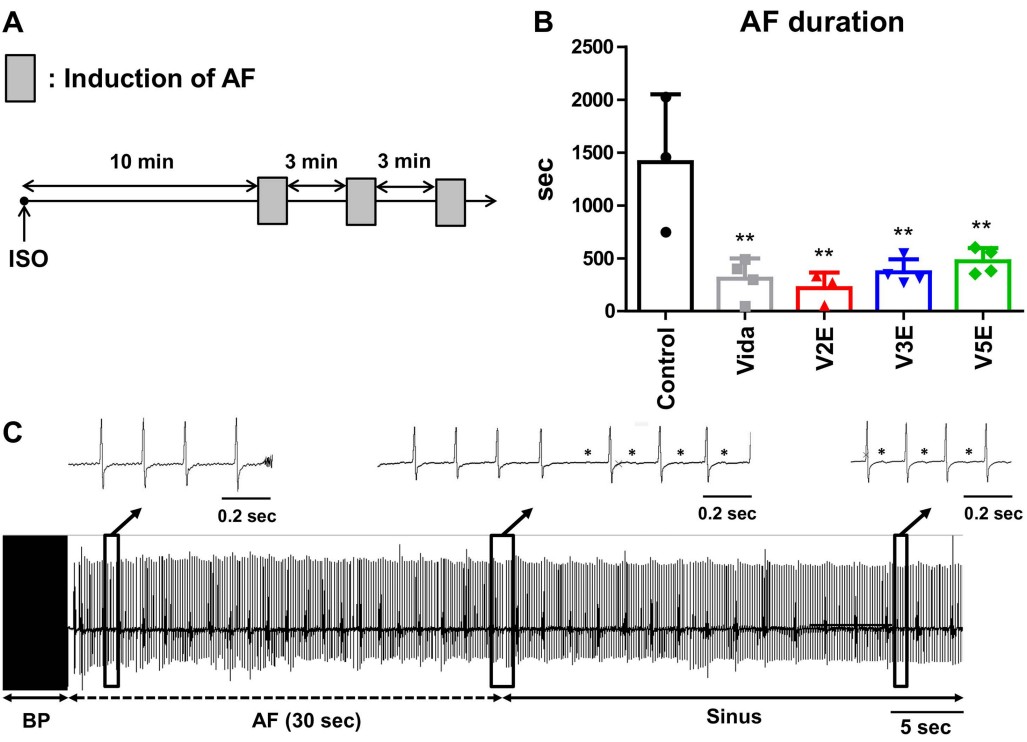

**Fig 7. Effects of vidarabine, V2E, V3E and V5E on ISO-induced AF.** (A) Schematic illustration of the experimental protocol for AF induction after sympathetic activation in mice. WT mice were given vidarabine (Vida) (15 mg/kg/day), V2E (19.7 mg/kg/day), V3E (19.7 mg/kg/day) or V5E (19.7 mg/kg/day) via subcutaneously implanted osmotic mini-pumps for 6 days. Ten minutes after intraperitoneal administration of ISO (1.5 mg/kg), AF was induced by transesophageal atrial burst pacing and the AF duration was measured. The rectangle represents the period from the start of burst pacing to the termination of AF. Note that the longest duration among 3 trials was taken to be the duration of AF for each individual animal. (B) Vidarabine, V2E, V3E and V5E significantly shortened the duration of ISO-elongated AF. Statistical significance was analyzed by one-way ANOVA with the Tukey-Kramer's *post hoc* test. **$P<0.01$ vs control (without inhibitor). Data are presented as mean±SD and dots show individual data from three to four WT mice (Control: $n=3$, Vida: $n=4$, V2E: $n=3$, V3E: $n=4$, V5E: $n=4$). (C) Representative surface ECG recordings of V2E-treated mice after induction of AF by atrial burst pacing. Asterisks indicate P-waves. BP: burst pacing.

These data indicate that V2E, V3E and V5E suppress the initiation and maintenance of AF after transesophageal rapid atrial pacing, probably through inhibition of AC5 activity in the atrium.

**V2E ameliorates dysregulation of $Ca^{2+}$ homeostasis after chronic ISO**

Since phosphorylation of most $Ca^{2+}$-handling proteins is altered in many models of experimental HF and AF, which might lead to increased $Ca^{2+}$ release from the sarcoplasmic reticulumn and disruption of $Ca^{2+}$ homeostasis, we examined the effects of vidarabine and V2E on protein kinase A-dependent ryanodine receptor 2 (RyR2) phosphorylation (Ser-2808). RyR2 phosphorylation at Ser-2808 was significantly increased after chronic ISO infusion (control vs. ISO: $100 \pm 16\%$ vs. $224 \pm 42\%$, $P < 0.001$). However, this increase was significantly suppressed in vidarabine- or V2E-treated mice (vidarabine: $161 \pm 21\%$, $P < 0.05$ vs ISO; V2E: $140 \pm 20\%$, $P < 0.05$ vs ISO) (S12A Fig of S1 Data).

Calpain is an intracellular $Ca^{2+}$-activated protease and an important mediator of the action of calcium. Calpains are present predominantly in their full-length, unautolysed/unactivated forms at rest. However, calpain appears in autolyzed forms upon disruption of $Ca^{2+}$-homeostasis during cardiac pathogenesis such as HF, AF, hypertrophy or ischemic reperfusion [33,42]. We thus examined calpain 1 activity by measuring the level of autolyzed calpain 1 in cardiac tissue and found that calpain 1 autolysis was significantly greater after chronic ISO infusion (control vs ISO: $100 \pm 4\%$ vs $145 \pm 15\%$, $P < 0.001$). However, this increase was significantly inhibited in the heart of vidarabine- or V2E-treated mice (vidarabine: $118 \pm 5\%$, $P < 0.05$ vs ISO; V2E: $120 \pm 19\%$, $P < 0.05$ vs ISO) (S12B Fig of S1 Data).

These data suggest that vidarabine and V2E protect the heart from catecholamine-mediated dysregulation of $Ca^{2+}$ homeostasis to comparable extents.

## Discussion

HF affects approximately 64 million people worldwide with high rates of hospitalization and mortality [43], and its prevalence is projected to increase due to the aging of the population [44]. Importantly, HF patients in low-income and middle-income countries are at a higher risk of death compared with those in high-income countries [45]. Therefore, effective, safe, and low-cost treatments, such as medication, that do not require expensive medical device would be helpful to combat HF. Due to the fact that adverse side-effects of β-blockers, such as worsening heart function and respiratory comorbidities, significantly limits the usefulness of these agents, there is still an urgent need for the development of safe and effective drugs for HF therapy.

We have established the efficacy of vidarabine itself on the pathogenesis of CVD in animal models [6,13,46–48]. However, the potential usefulness of vidarabine is limited by its poor solubility, which results in a requirement for prolonged infusion in relatively large volumes of intravenous fluid. Furthermore, vidarabine is deaminated in the gastrointestinal tract by ADA to Ara-H, which was found to possess little inhibitory effects on AC activity in this study.

The introduction of a polar functional group is an established approach to increase water solubility [49]. For example, introduction of a (dimethylamino)acetyl group is effective to improve water solubility [50], and may improve intestinal absorption as well. Watabe *et al*. recently reported that submission of a (dimetylamino)acetyl group into α-tocotorienol (an *N*,*N*-dimethylglycine ester prodrug) improves intestinal absorption as a result of self-micellization with intrinsic bile acid [51].

Therefore, we set out to develop vidarabine derivatives with improved solubility by means of (dimethylamino)acetic acid substitution at the 2'-, 3'- or 5'- position of the arabinose ring (V2E, V3E and V5E respectively). Evaluation of these compounds *in vitro* and *in vivo* demonstrated that V2E, V3E and V5E indeed possess higher water solubility than vidarabine, and further, V3E is resistant to ADA. The inhibitory effect of these compounds on cardiac AC activity was comparable to that of vidarabine. In addition, animal experiments revealed that V2E, V3E and V5E ameliorated cardiac dysfunction and AF susceptibility.

Among the nine mammalian isoforms of AC subtypes, AC6 is the major fetal cardiac isoform, and AC5 is the major cardiac isoform in adults [52–54]. Disruption of the AC5 gene has no effect on baseline cardiac function, but leads to an increase in heart rate (HR) despite reduced baseline AC activity. The increased basal HR may be related to a loss of parasympathetic restraint, because AC5 is a Gi-inhibitable AC isoform, although other so-far-unidentified mechanisms may also play a role in regulating baseline cardiac function and HR in AC5KO [12]. AC5KO mice exhibit significant resistance to various stresses on the heart, such as chronic catecholamine infusion [10], pressure overload [55], and aging [14]. The AC5KO model is also resistant to the development of diabetes and obesity, as well as cardiomyopathy induced by aging and diabetes, and exhibits longevity and improved exercise capacity [56,57]. All of these effects are mediated, in part by protection against oxidative stress [56]. Furthermore, the present study showed that the ISO-induced prolongation of AF was significantly blunted in AC5KO mice as compared to control mice (S3A Fig of S1 Data).

In contrast to AC5, several studies have shown that an increase in AC6 expression is beneficial for the failing heart, preserving LV contractile function and reducing dilation and dysfunction in hearts showing pressure overload [58,59]. AC6 overexpression prevents cardiac hypertrophy, fibrosis, and cardiomyopathy [60,61]. Indeed, AC6KO mice exhibit reduced cAMP production and have significantly higher mortality compared to AC6WT. Overexpression of AC8 in the heart has no detrimental consequence on global cardiac function, despite a 7-fold increase in basal AC activity and a 4-fold increase in protein kinase A activity in the heart [62].

Therefore, AC5 appears to have pharmacological potential, because suppressing the activity of AC5, and not the entire β-AR signaling pathway, may be preferable to current β-AR blockade therapy for the treatment of HF and arrhythmia.

ADA is a cytosolic enzyme that participates in purine metabolism, degrading adenosine or 2'-deoxyadenosine to inosine or 2'-deoxyinosine, respectively [63]. Further metabolism of these deaminated nucleotides leads to hypoxanthine [63]. ADA also deaminates vidarabine to Ara-H via a similar mechanism [64]. Ara-H possesses some antiviral activity, but is at least 10-fold less potent than vidarabine [64], so this metabolic reaction is a major limitation in the use of vidarabine for antiviral therapy [21,65].

Absorption of vidarabine from the intestine is very poor, because vidarabine is highly susceptible to deamination by intestinal ADA. Thus, vidarabine should be given intravenously in order to achieve sufficiently high plasma levels. However, due to its poor water solubility, vidarabine must be administered in a large volume of an appropriate intravenous infusion fluid (e.g., glucose 5%). The recommended intravenous injection dose is 10–15 mg/kg daily, and this is slowly infused over a 12–24 hours period. Thus, new vidarabine derivatives are needed to overcome the biopharmaceutical limitations of vidarabine itself.

In this study, we demonstrated that the inhibitory effects of vidarabine, V2E, V3E and V5E on forskolin-stimulated cardiac AC activity were significantly decreased in the presence of ADA. However, the decrease of the inhibitory effects of V2E, V3E and V5E was smaller than that of vidarabine. In particular, V3E was relatively resistant to deamination by ADA, suggesting that V3E might be available for oral administration.

Our previous studies of AC5KO revealed that disruption of AC5 did not alter cardiac function at baseline and protected the heart from stresses [10,55]. AC5KO also showed increased longevity [13,14]. We have recently demonstrated that inhibition of AC5 with vidarabine attenuates adrenergic receptor stimulation-induced $Ca^{2+}$ leakage and spontaneous $Ca^{2+}$ release from sarcoplasmic reticulum. It also attenuates sympathetic activation-induced reactive oxygen species production in isolated cardiac myocytes [6], which is involved in various physiological and pathological processes in the heart, including fibrosis, apoptosis and cardiac dysfunction [13]. In the present study, V2E, V3E and V5E selectively inhibited AC5 over other AC subtypes expressed in the heart. This suggests that V2E, V3E and V5E might protect the heart from stress and prevent the progression of HF and susceptibility to AF in a similar manner to vidarabine.

It is well established that renin secretion by the juxtaglomerular cells (JG cells), in contrast to almost all secretary cells, is inversely related to intracellular calcium concentration [66,67]. Thus, paradoxically, elevated

intracellular calcium is a potent inhibitor of renin release [68]. AC5, which is a calcium-inhibitable AC isoform and is expressed in JG cells, was demonstrated to be the key enzymatic mediator of cAMP stimulation of renin secretion in response to decreased intracellular calcium concentration [68]. More importantly, the selective AC5 inhibitor NKY80, which is also a vidarabine derivative, completely blocks the increase of both cAMP content and renin release in isolated JG cells [69]. Our data, together with previous findings, suggest that the renin angiotensin system might be, at least to some degree, a therapeutic target of V2E, V3E and V5E for the treatment of CVD. More importantly, V2E, V3E and V5E are more soluble, and should not require prolonged infusion of large volumes of intravenous fluid. These compounds appear to be promising candidates for overcoming the limitations of vidarabine in CVD treatments.

In summary, our aim in this work was to discover derivatives that might be better tolerated by patients than vidarabine itself. V2E, V3E and V5E appear to be promising candidates to protect the heart from catecholamine-induced HF and AF, which frequently coexist and are associated with cardioembolic stroke, impaired quality of life, and increased mortality [70]. Thus, our findings, together with the previous studies of AC5KO, suggest that V2E, V3E and V5E might contribute the increase of healthy life expectancy, especially in aged patients.

Accordingly, the next steps will be to investigate the long-term effects of these vidarabine derivatives to compare them with existing therapies and to conduct clinical trials. In addition, a careful evaluation of the safety and effectiveness of the vidarabine derivatives in patients with respiratory diseases and aged patients will be needed.

## Limitations

A possible limitation in this study is that the TUNEL study was performed using cultured rat neonatal cardiac myocytes (Fig 6). We have previously demonstrated that vidarabine inhibits ISO-mediated cardiac myocyte apoptosis in both cultured rat neonatal cardiac myocytes [71] and cultured mouse neonatal myocytes [72]. However, further validations using neonatal mouse cardiomyocytes might be desirable in the future.

## Supporting information

**S1 Data.  Revision_S1 Data revised-2-add RS correction-02.**
(PDF)

**S2 Data.  S2_raw_images_original picture revised-02.**
(PDF)

**S3 Data.  Data.**
(XLSX)

## Author contributions

**Conceptualization:** Kenji Suita, Yujiro Hoshino, Yoshihiro Ishikawa, Satoshi Okumura.

**Formal analysis:** Kenji Suita, Yoshio Hayakawa, Yujiro Hoshino, Wenqian Cai, Reiko Kurotani, Yoshiki Ohnuki, Yasumasa Mototani, Yoshihiro Ishikawa, Satoshi Okumura.

**Funding acquisition:** Kenji Suita, Yoshiki Ohnuki, Yoshihiro Ishikawa, Satoshi Okumura.

**Investigation:** Kenji Suita, Yoshio Hayakawa, Wenqian Cai, Reiko Kurotani.

**Methodology:** Kenji Suita, Yujiro Hoshino, Wenqian Cai, Reiko Kurotani, Satoshi Okumura.

**Supervision:** Yoshihiro Ishikawa, Satoshi Okumura.

**Writing – original draft:** Kenji Suita, Yujiro Hoshino, Satoshi Okumura.

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
