## [Decision Letter · Decision Letter 0]

2 Oct 2024

Dear Dr. Okumura

Thank you for submitting your manuscript to PLOS ONE. After careful consideration, we feel that it has merit but does not fully meet PLOS ONE’s publication criteria as it currently stands. Therefore, we invite you to submit a revised version of the manuscript that addresses the points raised during the review process.

We look forward to receiving your revised manuscript.

Kind regards,

Vishal Chavda, MS, PhD, FCNIP

Academic Editor

PLOS ONE

Journal Requirements:

**Additional Editor Comments:**

Dear Authors,

Thank you for submitting your manuscript to PLOS ONE. WE have completed the review process and received expert suggestions from the peers. After my careful evaluation and reviewers suggestions, your manuscript required major revisions. Please complete revisions on time and re-submit.

Best,

Reviewers' comments:

Reviewer's Responses to Questions

**Comments to the Author**

1. Is the manuscript technically sound, and do the data support the conclusions?

Reviewer #1: Yes

Reviewer #2: Yes

Reviewer #3: Partly

2. Has the statistical analysis been performed appropriately and rigorously?

Reviewer #1: Yes

Reviewer #2: I Don't Know

Reviewer #3: Yes

3. Have the authors made all data underlying the findings in their manuscript fully available?

Reviewer #1: Yes

Reviewer #2: Yes

Reviewer #3: Yes

4. Is the manuscript presented in an intelligible fashion and written in standard English?

Reviewer #1: No

Reviewer #2: Yes

Reviewer #3: Yes

Reviewer #1: The manuscript appears to be well-written in terms of technical language and clarity. However, there are a few areas where slight improvements can enhance readability:

1. Sentence Structure: Some sentences are quite long and could be split for clarity. Shorter sentences can make the text easier to follow.

2. Consistency: Ensure that terms and abbreviations are used consistently throughout the manuscript. For example, if you introduce a term like "AC5" or "ADA," make sure it is consistently abbreviated after the first mention.

3. Punctuation: Review for proper punctuation, especially in complex sentences. Commas can help clarify meaning and improve flow.

4. Technical Jargon: While the audience may be familiar with specific terms, consider briefly defining less common jargon or acronyms when first introduced.

5. Transitions: Use transitional phrases to improve the flow between sections and ideas, helping guide the reader through the results and discussions.

Reviewer #2: Authors tackled to prevent from heart failure and arrhythmia by specifically targeting adenylyl cyclase isoform, while avoiding cardiac function suppression. First, they synthesized derivatives (V2E, V3E and V5E) of vidarabine, by the substitution 2’, 3’ and 5’ position of arabinose with a (dimethylamino) acetic acid group, to improve its poor solubility. Then confirmed an increased solubility of V2E, V3E and V5E compared to vidarabine. Further experimentally showed V2E, V3E and V5E reduced the 1) development of heart failure and 2) susceptibility to atrial fibrillation using mouse model.

Overall idea is interesting, and approach is proper. However, I believe current manuscript needs visual improvement (type of graphs) and clarifications, including following points /questions/comments.

Introduction.

1. Authors claimed that “AC5 is expressed specifically in the heart, with little expression in the lung [5]”. However, according to cited literature, AC5 is detected in the heart at comparable level to brain. Therefore, authors either modify the above-mentioned claim or include additional reference to support AC5 expression is specific to heart. Otherwise, have a discussion on this point.

Method section.

2. Even though it is understandable from the study, authors should clearly describe animal species that was used in the study, otherwise it is unclear from their “animals” subsection.

3. Also, please include number of animals used in the study per group per experiment, as well as their sex.

4. Authors missing subsection for “cell culture” that describes cell line including its

a. name, if it is primary culture cells then

b. isolation procedure,

c. cell culture conditions including media supplements, passage number and

d. seeding density etc.

5. Please clarify why 30oC was chosen to conduct ADA mediated deamination study when ADA and V2E, V3E or V5E mixture was incubated for 5 min.

6. AF induction, why authors waited 10 minutes after IP administration of ISO at 15mg/kg? how this concentration was determined? (if adopted, needs a ref).

7. Measurement of fibrosis area in heart tissue, it is not clear for what was the size/field (as well as its unit) of tissue/images were analyzed to obtain AF area after Masson trichrome staining.

Result section.

In general, according to figure legend each dot in the graph represents individual data. However, it is not clear why some group has more individual data points than other group. Also, consistency of graph is important for the readers, why some example Fig 4A is bar graph with error bar while showing no individual data points, but Fig 4B shows individual data points. Please clearly state how many animals were used in that experiment for each figure (except cell culture study), because it is not clear if individual data points are repeated measurements from single animal or each dot represents different animals.

8. What was the half-life of V2E, V3E and V5E?

9. Authors should consider finding better way to present Figure 2-4.

10. Fig 5. For inserted scale bar, it’s unit is not described in the figure legend. As mentioned before, it is critical to include number of animal/ size of tissues analyzed for that experiment. Provided image does not clearly show dapi positive cells.

11. Fig 6. TUNEL assay, perhaps it is more informative to plot “number of positive stained cells/dapi positive cell” rather than “Tunel positive (%)” on the Y axis of the graph. Please add scale bar to the figure, if possible.

Discussion.

12. Eventhough, authors claimed that “AC5KO revealed that genetic disruption of AC5 did not affect cardiac function at baseline [7]”, however their cited study indicated that conscious AC5KO mice showed significantly increased heart rate compared to WT. Therefore, it would be critical to discuss how current result could be affected by this.

13. Authors should discuss why rat neonatal cardiomyocytes were used for apoptotic assay (TUNEL), over mouse derived cardiomyocytes. If not, it should be noted as their experimental limitation(s).

Reference.

14. References are not uniformly presented such as .[X] rather than [X].

Author contribution.

15. Could you please add name of author who conducted statistical analysis to “Author contribution” section under “statistical analysis” subsection?

Reviewer #3: Authors do not provide enough validation of their synthesized compounds. They provide is a molecular weight for each of the compounds. It is recommended authors should include percent yields, proton NMR, and some sort of mass spectrometry trace, be it from an LC-MS, HPLC, or hi-res mass spectrometer. These are all standard forms of validation for synthesized compounds, and therefore should be reasonable expectations for the manuscript.

Cardiac hypertrophy accompanies many forms of heart disease, including ischemic disease, hypertension, heart failure, and valvular disease. ventricular hypertrophy is associated with significantly increased risk of heart failure. In this context cellular calcium homeostasis also plays crucial role. Hence, it is recommended to check the effect of the vidarabine derivative on calcium homeostasis and cardiac hypertrophy.

It is well known that the isoform AC5 deletion, results in protecting the heart from cardiomyopathy, chronic catecholamine stress, and chronic pressure overload. On the contrary, the overexpression of isoform AC6 leads to increased LV function, improved cAMP and Ca2+ handling, and may protect the heart from pressure overload-induced systolic and diastolic dysfunction. The author should check the effect of different vidarabine derivatives on the activity of AC6 as well.

Numerous profibrotic factors have been identified at the molecular level such as TGFβ, IL11, AngII, which activate gene expression programs for myofibroblast activation. Therapeutic inhibition of profibrotic factor, which is the master‐regulator of fibrosis whether mediated by the vidarabine derivative would be significant finding.

How Author has determined the dose of Vidarabine derivatives

**Do you want your identity to be public for this peer review?** For information about this choice, including consent withdrawal, please see our Privacy Policy

Reviewer #1: **Yes: ** Sara Abou Al-Saud

Reviewer #2: No

Reviewer #3: **Yes: ** Rima Chattopadhyay

---

## [Author Response · Author response to Decision Letter 1]

29 Mar 2025

Here are some General comments on the manuscript:

Criticism-1:

The introduction effectively sets the stage for the study by highlighting the importance of heart failure and the role of the autonomic nervous system. However, some sentences are long and complex, which may make it harder for reader to follow. Consider breaking them into shorter, more digestible sentences.

Response-1:

We have reviewed the whole manuscript as you suggested, and have split up sentences that seemed particularly long or comlex.

Criticism-2:

While HF is mentioned as a leading cause of mortality, a brief explanation of its prevalence, types and associated symptoms would provide readers with essential background. This will help underscore the urgency of developing new treatments.

Response-2:

Thank you for your suggestion. We incorporated the following sentences in the discussion of the revised manuscript (Page 32, Lines 2-8).

HF affects approximately 64 million people worldwide with high rates of hospitalization and mortality [1], and its prevalence is projected to increase due to the aging of the population [2]. Importantly, HF patients in low-income and middle-income countries are at a higher risk of death compared with those in high-income countries [3]. Therefore, effective, safe, and low-cost treatments, such as medication, that do not require expensive medical devices would be helpful to combat HF.

Criticism-3:

The statement about the autonomic nervous system could benefit from clarification. Specify how its long-term activation adversely affects cardiac function. Including a brief description of the sympathetic nervous system’s role would help.

Response-3:

We incorporated the following sentences in the introduction of the revised manuscript (Page 5, Lines 3-11).

Under physiological conditions, the sympathetic nervous system supports cardiac activity through the modulation of dromotropy, cronotropy, inotropy, and lusinotropy. Moreover, the balance between the sympathetic and parasympathetic nervous systems regulates the peripheral resistance and cardiac output, and plays an essential role in the regulation of cardiovascular function in response to stresses [4]. However, long-term activation of the sympathetic nervous system, characterized by increased release of norepinephrine and epinephrine from both the chromaffin cells of the adrenal gland and heart sympathetic fibers, has a deleterious effect on cardiac structure and performance, leading to progression of HF and arrhythmia [5].

Criticism-4:

When discussing β-blockers, it would be helpful to explain how they work and why their adverse effects are particularly problematic in older patients. For example, they can mention how β-blockers manage heart rate and contractility but can lead to complications in patients with coexisting pulmonary conditions.

Response-4:

We incorporated the following sentences in the introduction of the revised manuscript (Page 6, Lines 5-8)

Since β-ARs are also expressed in the pulmonary bronchus and in cardiac myocytes, β-blockers can cause bronchial smooth muscle contraction and altered cardiac function, resulting in abnormalities of respiratory function and the cardiac circulatory system.

Criticism-5:

When introducing AC5, its specificity to the heart was mentioned but an elaboration on its role in cardiac function and how it differs from other isoforms is important. This will enhance understanding of why selective inhibition of AC5 is a viable therapeutic strategy.

Response-5:

We incorporated the following sentences in the discussion of the revised manuscript (Page 34, Line 12-Page 35, Line 2)

In contrast to AC5, several studies have shown that an increase in AC6 expression is beneficial for the failing heart, preserving LV contractile function and reducing dilation and dysfunction in hearts showing pressure overload [6, 7]. AC6 overexpression prevents cardiac hypertrophy, fibrosis, and cardiomyopathy [8, 9]. Indeed, AC6KO mice exhibit reduced cAMP production and have significantly higher mortality compared to AC6WT. Overexpression of AC8 in the heart has no detrimental consequence on global cardiac function, despite a 7-fold increase in basal AC activity and a 4-fold increase in protein kinase A activity in the heart [10]. .

Criticism-6:

The discussion about vidarabine’s solubility issues is critical. Consider including specific examples of how poor solubility impacts clinical administration and patients outcomes, emphasizing the need for improved formulations.

Response-6:

We incorporated the following sentences in the discussion of the revised manuscript (Page 35, Line 13-Page 35, Line 2)

Absorption of vidarabine from the intestine is very poor, because vidarabine is highly susceptible to deamination by intestinal ADA. Thus, vidarabine should be given intravenously in order to achieve sufficiently high plasma levels. However, due to its poor water solubility, vidarabine must be administered in a large volume of an appropriate intravenous infusion fluid (e.g., glucose 5%). The recommended intravenous injection dose is 10-15 mg/kg daily, and this is slowly infused over a 12-24 hours period. Thus, new vidarabine derivatives are needed to overcome the biopharmaceutical limitations of vidarabine itself.

Criticism-7:

When stating that direct inhibition of AC could prevent pathological β-AR activation, briefly mention the potential physiological benefits of this approach. This could include improved cardiac function or reduced arrhythmic risk.

Response-7:

We incorporated the following sentences in the results portion of the revised manuscript (Page 29, Lines 10-14).

---, because suppressing the activity of AC5, and not the entire β-adrenergic signaling pathway, may be preferable to current β-AR blockade therapy for the treatment of heart failure and arrhythmia. Specifically, it is expected to reduce the risk of adverse effects on cardiac function and the respiratory tract, especially in aged patients.

Criticism-8:

Previous studies on AC5KO mice were mentioned. Summarizing key findings from these studies (e.g., effects on longevity and stress resistance) will strengthen the rationale for targeting AC5 in the current work.

Response-8:

We incorporated the following sentences in the discussion portion of the revised manuscript (Page 34, Lines 5-8).

The AC5KO model is also resistant to the development of diabetes and obesity, as well as cardiomyopathy induced by aging and diabetes, and exhibits longevity and improved exercise capacity [11, 12]. All of these effects are mediated, in part by protection against oxidative stress [11].

Criticism-9:

It was noticed that the transition to discussing vidarabine derivatives is abrupt. Consider a bridging sentence that connects the limitations of vidarabine to the need for developing these new derivatives. Explain how modifying the structure could overcome previous obstacles.

We incorporated the following sentences in the introduction of the revised manuscript (Page 9, Lines 1-8).

Several vidarabine derivatives in which the 2’-, 3’- or 5’-OH group of the arabinose ring is converted to a more polar functional group, such as phosphate [13], amino acid ester [14], or ester [15], show increased water solubility and resistance to deamination by ADA. More importantly, we revealed that vidarabine possesses a pharmacophore structure for AC5 inhibition at the C2 and C6 positions (=C2H-N1 = C6(NH2)-) of the adenosine ring [16]. Therefore, modification at the arabinose ring of vidarabine was thought to be a promising approach for overcoming vidarabine’s limitations without loss of the AC5-inhibitory effect.

Criticism-10:

In the objectives of the study, the final paragraph outlines the focus on vidarabine derivatives but could more explicitly state the main objectives of the study. Clearly define what you aim to achieve with this research, such as improving solubility and reducing side effects.

Response-10:

We incorporated the following sentences in the discussion of the revised manuscript (Page 37, Lines 17-Page 38, Line 5). Please note that the preceding sentences already mention the potential advantage of V2E, V3E and V5E, i.e., that they should not require prolonged infusion of large volumes of intravenous fluid due to their increased solubility.

In summary, our aim in this work was to discover derivatives that might be better tolerated by patients than vidarabine itself. V2E, V3E and V5E appear to be promising candidates to protect the heart from catecholamine-induced HF and AF, which frequently coexist and are associated with cardioembolic stroke, impaired quality of life, and increased mortality [17]. Thus, our findings, together with the previous studies of AC5KO, suggest that V2E, V3E and V5E might contribute the increase of healthy life expectancy, especially in aged patients.

Here are some Specific comments on the manuscript:

Criticism-1:

The statistic stating that HF affects “an estimated 26 million people and contributes to more than 1 million hospitalizations” could be enhanced by citing specific sources. Ensure that the references are up-to-date and relevant.

Response-1:

We have addressed this. Please see our response to criticism-2 of the general comments.

Criticism-2:

The mention of β-blockers and their limitations is important, but it might be beneficial to briefly explain how they currently work in HF management before discussing their adverse effects.

Response-2:

We incorporated the following sentences in the introduction of the revised manuscript (Page 5, Lines 15-17).

Controlled clinical trials have shown that β-blockers can reduce the risk of death as well as the risk of hospitalization for cardiovascular causes in patients with heart failure [18].

Criticism-3:

The discussion on the synthesis of vidarabine derivatives (V2E, V3E, V5E) could be more detailed. Providing some background on how these specific modifications improve solubility would be helpful.

Response-3:

We incorporated the following sentences in the discussion of the revised manuscript (Page 33, Lines 1-4).

The introduction of a polar functional group is an established approach to increase water-solubility [19]. For example, introduction of a (dimethylamino)acetyl group is effective to improve water solubility [20], and may improve intestinal absorpotion as well [21, 22].

Criticism-4:

The findings on the efficiency of V2E, V3E, and V5E are significant, however, adding a comparison to existing therapies could help emphasize their potential advantages.

Response-4:

We incorporated the following sentences in the introduction of the revised manuscript (Page 6, Line 12-Page 7, line 4)

Approximately 30% of cardiac surgical patients develop postoperative atrial fibrillation (POAF) [23]. POAF significantly increases the duration of postoperative hospital stay, hospital costs, and the risk of recurrent AF. Moreover, POAF has been associated with a variety of adverse cardiovascular events such as stroke, heart failure, and mortality. Perioperative β-blockers are the mainstay for POAF prophylaxis in patients undergoing cardiac surgery, as recommended by international guidelines (Class of Recommendation I, Level of Evidence A in the 2020 European Society of Cardiology [ESC] guidelines for the diagnosis and management of AF) [24]. However, their inhibitory effects on cardiac function and respiration limit the dosage and duration in patients after cardiac surgery. Consequently, there is a need for new β-AR blockade therapy without these adverse effects on cardiac function and respiration.

Criticism-5:

The section on AC isoforms (AC5 and AC6) is informative. Consider summarizing the key implications of the findings related to these isoforms for a broader audience who may not be familiar with this area.

Response-5:

We have added the requested information. Please see our response to criticism-5 of the general comments.

Criticism-6:

When discussing how V2E, V3E, and V5E selectively inhibit AC5, it would be helpful to provide some mechanistic insights. A diagram could be useful to illustrate these interactions and effects.

Response-6:

The pharmacophore structure at the C2 and the C6 positions (=C2H-N1=C6(NH2)-) of the adenine ring is essential for the selective inhibition of AC5, and previous P-site inhibitors with a selective inhibitory effect on AC5, including vidarabine, contains this pharmacophore (Fig 1A) [16]. V2E, V3E, V5E also contain this pharmacophore structure and thus show the selective inhibition on AC5.

We modified Fig 1 to frame the pharmacophore (=C2H-N1=C6(NH2)-) with a red-dotted rectangle, which should clarify the situation for readers (see revised Fig 1).

In addition, we incorporated the following sentences in the Fig 1 legend of the revised manuscript (Page 56, Lines 5-6).

The essential pharmacophore structure (=C2H-N1=C6(NH2)-) for selective inhibition of AC5 is framed with a red-dotted rectangle

Criticism-7:

Referencing figures consistently throughout the results aids in visual comprehension and supports the textual claims. Ensure that the figures are clearly labelled and match the referenced data.

Response-7:

We re-checked and confirmed that the figures are clearly labelled and match the referenced data.

Criticism-8:

The findings on fibrosis and apoptosis are well-articulated. Including a brief discussion on the underlying mechanisms by which V2E, V3E, and V5E exert their protective effects could add depth to your results.

Response-8:

We have added the requested information. Please see our response to criticism-4 from Reviewer#3.

Criticism-9:

If possible, use tables to summarize key data points, such as solubility levels, AC activity measurements, and cardiac function parameters to improve readability and quick reference for readers.

Response-9:

As suggested, we have summarized the key data of vidarabine derivatives (water solubility, AC5 selectivity, resistance to ADA, anti-heart failure effect and anti-AF effect) in Table 1 in the revised manuscript.

Criticism-10:

Explain how poor solubility impacts clinical use, such as the need for intravenous administration and associated complications like fluid overload. This highlights the urgency for improved formulations.

Response-10:

Done as requested. Please see our response to criticism-6 of the general comments.

Criticism-11:

Explain why substituting with (dimethylamino)acetic acid at specific positions enhances solubility. A brief description of the chemical properties that affect solubility could clarify this for readers not familiar with medical chemistry.

Response-11:

We incorporated the following sentences in the discussion of the revised manuscript (Page 33,Lines 4-6).

Watase et al. recently suggested that substituation of a (dimethylamino)acetyl group into α-tocotorienol (an N,N-dimethylglycine ester prodrug) improves intestinal absorption as a result of self-micellization with intrinsic bile acid [21].

Criticism-12:

End with a paragraph discussing potential future studies. This could include clinical trials, investigations into long-term effects, or comparisons with existing therapies. Highlight any specific patient population that may benefit most from these new compounds.

Response-12:

We incorporated the following sentences in the discussion section of the revised manuscript (Page 38, Lines 6-9).

Accordingly, the next step will be to investigate the long-term effects of these vidarabine derivatives to compare them with existing therapies and to conduct clinical trials. In addition, a careful evaluation of the safety and effectiveness of the vidarabine derivatives in patients with respiratory diseases and aged patients will be needed.

References

1. Shahim B, Kapelios CJ, Savarese G, Lund LH. Global public health burden of heart failure: An updated review. Card Fail Rev. 2023;9:e11. https://doi:10.15420/cfr.2023.05. PMID: 37547123.

2. Panisello-Tafalla A, Haro-Montoya M, Caballol-Angelats R, Montelongo-Sol M, Rodriguez-Carralero Y, Lucas-Noll J, et al. Prognostic significance of lung ultrasound for heart failure patient management in primary care: A systematic review. J Clin Med. 2024;13(9)2460. h

---

## [Decision Letter · Decision Letter 1]

25 Jun 2025

Dear Dr. Okumura,

There is a minor request by one of reviewers that I consider is easy to meet and helps on data transparency. Please,provide original pictures for all western blots as supplementary material.

We look forward to receiving your revised manuscript.

Kind regards,

Agustín Guerrero-Hernandez

Academic Editor

PLOS ONE

Journal Requirements:

Reviewers' comments:

Reviewer's Responses to Questions

**Comments to the Author**

Reviewer #2: All comments have been addressed

Reviewer #3: All comments have been addressed

2. Is the manuscript technically sound, and do the data support the conclusions?

Reviewer #2: Yes

Reviewer #3: Yes

3. Has the statistical analysis been performed appropriately and rigorously?

Reviewer #2: Yes

Reviewer #3: Yes

4. Have the authors made all data underlying the findings in their manuscript fully available?

Reviewer #2: No

Reviewer #3: Yes

5. Is the manuscript presented in an intelligible fashion and written in standard English?

Reviewer #2: Yes

Reviewer #3: Yes

Reviewer #2: Thanks for addressing the raised comment/questions.

A few points from my side:

1) Authors should provide original pictures for all western blots these are shown in the supplementary figure (including molecular weight marker).

2) Please remove website link from the manuscript (page 18, at the end of subsection "measurement of fibrosis are in heart tissue").

Reviewer #3: All the questions asked are almost discussed and experiments also done in detailed manner except some which I could see asked by other reviewer probably beyond scope.

Explanation for all the criticism seems reasonable

**Do you want your identity to be public for this peer review?** For information about this choice, including consent withdrawal, please see our Privacy Policy

Reviewer #2: No

Reviewer #3: No

---

## [Author Response · Author response to Decision Letter 2]

7 Jul 2025

Reviewer #2:

Thanks for addressing the raised comment/questions.

A few points from my side:

1) Authors should provide original pictures for all western blots these are shown in the supplementary figure (including molecular weight marker).

Response: I rechecked and confirmed that original pictures for all western blots are provided in the S2 Data. However, original pictures did not contain molecular weight marker. We thus provide original pictures including molecular weight marker for all western blots (Supplementary figure 12 in S1 Data) in S2 Data.

2) Please remove website link from the manuscript (page 18, at the end of subsection “measurement of fibrosis are in heart tissue”).

Response: As requested, we have removed the website link.

Reviewer #3:

All the questions asked are almost discussed and experiments are done in detailed manner except some which I could see asked by other reviewer probably beyond scope. Explanation for all the criticism seems reasonable.

Response: Thank you.

---

## [Editor Report · Decision Letter 2]

4 Aug 2025

Water-soluble vidarabine derivatives alleviate catecholamine-induced heart failure and arrhythmia without impairing cardiac function in mice

PONE-D-24-27843R2

Dear Dr. Okumura,

We’re pleased to inform you that your manuscript has been judged scientifically suitable for publication and will be formally accepted for publication once it meets all outstanding technical requirements.

Kind regards,

Agustín Guerrero-Hernandez

Academic Editor

PLOS ONE
---

## [Editor Report · Acceptance letter]

PONE-D-24-27843R2

PLOS ONE

Dear Dr. Okumura,

I'm pleased to inform you that your manuscript has been deemed suitable for publication in PLOS ONE. Congratulations! Your manuscript is now being handed over to our production team.

Kind regards,

on behalf of

Dr. Agustín Guerrero-Hernandez

Academic Editor

PLOS ONE